# Genome-Wide SNP Markers Accelerate Perennial Forest Tree Breeding Rate for Disease Resistance through Marker-Assisted and Genome-Wide Selection

**DOI:** 10.3390/ijms232012315

**Published:** 2022-10-14

**Authors:** Mehdi Younessi-Hamzekhanlu, Oliver Gailing

**Affiliations:** 1Department of Forestry and Medicinal Plants, Ahar Faculty of Agriculture and Natural Resources, University of Tabriz, 29 Bahman Blvd., Tabriz P.O. Box 5166616471, Iran; 2Department of Forest Genetics and Forest Tree Breeding, University of Göttingen, Büsgenweg 2, D-37077 Göttingen, Germany

**Keywords:** disease resistance, QTL mapping, GWAS, genomic-wide selection

## Abstract

The ecological and economic importance of forest trees is evident and their survival is necessary to provide the raw materials needed for wood and paper industries, to preserve the diversity of associated animal and plant species, to protect water and soil, and to regulate climate. Forest trees are threatened by anthropogenic factors and biotic and abiotic stresses. Various diseases, including those caused by fungal pathogens, are one of the main threats to forest trees that lead to their dieback. Genomics and transcriptomics studies using next-generation sequencing (NGS) methods can help reveal the architecture of resistance to various diseases and exploit natural genetic diversity to select elite genotypes with high resistance to diseases. In the last two decades, QTL mapping studies led to the identification of QTLs related to disease resistance traits and gene families and transcription factors involved in them, including NB-LRR, WRKY, bZIP and MYB. On the other hand, due to the limitation of recombination events in traditional QTL mapping in families derived from bi-parental crosses, genome-wide association studies (GWAS) that are based on linkage disequilibrium (LD) in unstructured populations overcame these limitations and were able to narrow down QTLs to single genes through genotyping of many individuals using high-throughput markers. Association and QTL mapping studies, by identifying markers closely linked to the target trait, are the prerequisite for marker-assisted selection (MAS) and reduce the breeding period in perennial forest trees. The genomic selection (GS) method uses the information on all markers across the whole genome, regardless of their significance for development of a predictive model for the performance of individuals in relation to a specific trait. GS studies also increase gain per unit of time and dramatically increase the speed of breeding programs. This review article is focused on the progress achieved in the field of dissecting forest tree disease resistance architecture through GWAS and QTL mapping studies. Finally, the merit of methods such as GS in accelerating forest tree breeding programs is also discussed.

## 1. Introduction

Forests have provided a unique ecosystem for terrestrial life for millions of years and inimitable resources for humans and other animals and microorganisms. The economic importance of forests in most countries and regions of the world is undeniable, and many industries, including wood, construction, paper and other industries, are dependent on forest products. However, there are several factors that have disturbed the balance of this ecosystem and the sustainable use of these resources in different parts of the world [1]. These catastrophic destructive changes are taking place with increasing intensity in developing countries and arid regions of the world with a prolonged drought stress, which have also been affected by the negative effects of climate change [2].

Along with other factors that threaten forests, such as the development of different industries, urbanization, agriculture, road construction, etc., various diseases are also important factors that have an adverse impact on the survival of many forest trees, which ultimately can lead to rapid and widespread deforestation. In the last decades, several diseases and pests have spread widely in different forest regions at an increasing rate. These diseases, which are mainly caused by fungal pathogens, destroy—for example—ash trees and result in the decline of several forest tree species [3]. In some cases, these diseases can threaten the survival of a particular species in a large area. For example, ash dieback disease caused by the fungal pathogen *Hymenoscyphus fraxineus* has spread widely in different parts of Europe and destroyed ash trees (*Fraxinus excelsior* L.), and has resulted in dramatic damage to and decline of ash trees in temperate forests [4]. Over the past few decades, oak charcoal disease has become a major problem in the forests of Zagros and Arasbaran, Iran, and is spreading rapidly, posing a serious threat to the survival of these forests. The disease, which is caused by *Biscogniauxia mediterranea* and *Obolarina persica* fungi, appears with high intensity due to long-term droughts in the target areas and increases the mortality of oak trees [5,6]. Chestnut blight is another important disease that has become an important challenge in forest areas. The cause of this disease is the fungal pathogen *Cryphonectria parasitica*, which has devastated chestnut forests in the North American region and other forests across the world [7,8]. In addition, there are several other fungal pathogens, including *Austropuccinia psidii*, *Fusarium* spp., *Rhizoctonia* spp., and *Rhytisma* spp., which affect important forest tree species such as *Eucalyptus* spp., *Fagus sylvatica* L., *Picea abies* (L.) Karst., and *Acer velutinum* Boiss., respectively, and cause serious losses [9,10,11,12].

In general, management strategies to reduce the destructive effects of various diseases are based on monitoring the prevalence of the pathogen, and its eradication or reduction. To ensure the adoption of effective management plans, it is necessary to pay attention to climate change, since different pathogens prefer certain temperatures and humidity, and are affected by the seasonal change in these factors and the formation of newly favorable climatic conditions for disease agents. Climate change can accelerate the spread and severity of their pathogenicity [13]. On the other hand, developing new adapted forest trees with acceptable resistance against biotic stresses (e.g., fungal diseases) will be necessary in integrative management of forests infected by these diseases. Traditional breeding methods (introducing new varieties, crossing and hybridization, selection in successive generations, provenance tests) and modern methods (e.g., genetic engineering and gene editing) can be used to improve forest trees [14]. The issue of time in plant breeding, especially in breeding disease-resistant forest trees, and the propagation of resistant trees as soon as possible, is extremely important in order to compensate for economic and ecosystem damage in the shortest possible time [15]. Molecular markers are among the precision tools available to breeders that can be used to accurately and quickly evaluate the various genotypes of trees in the early stages of development. These methods play an important role in accelerating the breeding and selection procedure, which can be used in a variety of breeding programs saving time and costs [16]. Although several genetic studies have been conducted for model crops and trees, in most forest trees this data is not satisfying. In recent years, the use of next generation sequencing (NGS) methods has become a widely used approach to genotyping and evaluating the diversity and structure of tree populations [17,18]. NGS methods and their derived markers result in high-throughput and reproducible data as compared with conventional molecular markers such as RAPDs, SSRs, AFLPs, etc. In addition, genomic sequencing can generate valuable information about the origin, epigenetic alterations, and physiological changes of plants. These data will improve our insight into pathogen-tree interactions and underlying resistance pathways, which in turn will be useful in adopting effective management strategies [3].

Quantitative trait locus (QTL) or interval mapping and genome-wide association studies (GWAS) are two methods which exploit the existing recombinant and linkage disequilibrium incidence, respectively. However, due to the low recombination rate in families derived from bi-parental crosses, GWAS is a more capable method for understanding the genetic architecture of disease resistance mechanisms in comparison with interval mapping. These two methods can identify markers associated with the desired trait to use in marker-assisted selection (MAS) of superior genotypes. Genomic selection methods can be used for the development of prediction models and selection of elite genotypes with high disease resistance characteristics. This method is practicable for resistance traits which are controlled by a large number of genes. However, selection of each of the mentioned methods is dependent on various factors including type of population and complexity of traits [16,19].

Genetic diversity is the basis of plant breeding, and on the other hand, evaluating this diversity and establishing a suitable diversity panel using a variety of genetic methods is an essential step in any breeding program. The use of NGS based procedures such as RNA-seq has made it possible to both identify the diversity of gene expression patterns and find several single nucleotide polymorphism (SNP) markers even in species without a reference genome [20]. These findings not only help to better understand the interactions between different genes involved in adaptation responses to various biological and abiotic stresses, but also accelerate the process of various breeding programs. In this review article, we intend to first investigate the molecular mechanisms of response of forest trees to various diseases, especially fungal diseases. In the following, we will discuss how molecular markers, especially NGS derived high-throughput markers, can help accelerate the process of breeding programs through QTL mapping and GWAS. We will also point out the role of genomic selection (GS) in the selection of superior disease-resistant trees. Finally, it is expected that the new findings will help to better understand the relationship between pathogen and tree, and accelerate the breeding programs so that a set of superior individuals can be selected for use in breeding programs and the development of resistant genotypes.

## 2. Molecular Mechanisms of Tree Responses to Diseases

Resistance is actually defined as the ability to survive for a long time despite the threat of disease. Trees use a variety of mechanisms to prevent the progression of pathogen invasion. The first defense obstacle of trees against the invasion of pathogens is the presence of physicochemical barriers such as the outer bark, the cuticle surface of leaves and their lignified cell walls. Forest trees, like many other plants, can secrete a wide range of phytochemicals such as terpenoids (mono- and diterpenoids), alkaloids, and phenols (such as stilbene, lignans, flavonoids, tannins, and proanthocyanidins) to change pathogen-host interactions in favor of host trees [21]. Moreover, phytohormones such as jasmonic acid, salicylic acid, methyl jasmonate, and abscisic acid can increase pathogen resistance by activating enzymes involved in the biosynthetic pathways of other compounds or by causing physiological changes [22]. It has been shown that in trees infected with pathogens, the expression of several gene families changes. Many of these genes mainly encode enzymes that are involved in various biosynthetic pathways, including phenylpropanoid, terpenoid, carbon metabolism, and so on. These gene families have also been reported to encode a variety of proteins, including fungicides, detoxifiers, inhibitors, and pathogen-related proteins (PR) [23,24]. So far, several families of PR proteins have been identified whose aggregation increases with the onset of pathogens and pests as well as abiotic stresses. These proteins play an important role in natural defense (both directly by destroying the pathogen cell wall and indirectly by producing eliminators) and ultimately trigger defense responses by inducers such as salicylic acid, jasmonic acid, and ethylene. In addition, polygalacturonase inhibitory proteins inhibit the action of the pathogenicity factor polygalacturonase by preventing the destruction of the cell wall in trees and the entry of pathogens, and enzymes such as polyphenol oxidase catalyze the oxidation reaction of hydroxyphenol to quinone derivatives to increase resistance of trees to pathogens [23].

In addition to existing defense structures, trees are able to exhibit a variety of defense responses after pathogens are detected. In the first stage of infection detection, receptors located in cell membranes called pattern recognition receptors (PRRs) and wall-associated kinases (WAKs) detect pathogen-associated molecular patterns (PAMPs) and damage-associated molecular patterns (DAMPs), respectively [24,25]. The immunity resulting from such a process is known as the PAMPs-triggered immunity (PTI). In fact, PRRs (Figure 1a) detect pathogen-derived compounds such as fungal chitin, bacterial flagellin, and viral dsRNA, while WAKs detect cellular structures degraded by the enzymatic activity of pathogens. The external part of the PRR receptor possessing leucine-rich repeats (LRR) is responsible for detecting and binding to the extracellular ligand, while the cytoplasmic domain acts as a kinase, initiating signal transduction [26]. There are other receptors inside the cell that can detect effectors that cross the membrane. These receptors, which belong to the nucleotide binding site-LRR (NBS-LRR) family of receptors encoded by R genes, are able to detect these effectors (Figure 1b). This pathway results in effector-triggered immunity (ETI). There are generally three domains in NBS-LRR (NLR) receivers, each of which has a specific function. At their N terminal, either the N-terminal Toll/interleukin-1 receptor-like (TIR) or coiled-coil (CC) domain can be placed, which starts the signal transduction after the effector is detected by other domains. The middle domain is NB, and with the continuous replacement of nucleotides (ATP instead of ADP), a conformational change occurs at this receptor, releasing the end of the N-terminal to initiate the signal transduction. The C-terminal domain is the ligand binding site that contains the duplicate units of leucine [27].

The mitogen-activated protein kinase (MAPK) cascade (Figure 1c) is one of the important signaling pathways in stress response that is stimulated by various stresses and ultimately leads to adaptive responses and improved resistance to biotic and abiotic stresses. This cascade begins with the phosphorylation of MAPKKK by the kinase proteins of the receptors (i.e., PRRs, WAKs, LRR and NBS-LRR). Phosphorylated MAPKKK then phosphorylates MAPKK. Subsequently, phosphorylated MAPKK leads to MAPK phosphorylation. Finally, the phosphorylated MAPK can act as a kinase enzyme and phosphorylate and activate different cellular targets such as protein kinases and TFs, and after transfer to the cell nucleus, leads to the expression of different genes to produce adaptive and resistance responses [28].

Turbulence in calcium concentrations is a common occurrence during stresses. This change in concentration activates the signaling pathway, which eventually alters the expression profile of the genes to elicit a suitable resistance response. PTI and ETI, during pathogen invasion, can lead to increased cytosolic calcium concentrations. The increase in the concentration of calcium ions in the cytosol (Figure 1d) is perceived by several proteins, including Ca^2^C–dependent protein kinases (CDPKs), calmodulin (CaM), calmodulin-like protein (CML), and then several TFs, such as calmodulin-binding transcription activators (CAMTAs) and WRKY, and leads to the transcription of the relevant genes (e.g., NDR1, EDS1, and RPS2 and RPM1 ETI protein) in the nucleus [24,25].

In addition to the role of plant hormones in controlling the developmental stages of plants, they also play a pivotal role in the formation of adaptive responses induced under stress conditions. While some hormones are mainly involved in growth regulation (e.g., auxins, gibberellins, cytokinins), other hormones such as jasmonic acid (JA), ethylene (ET), salicylic acid (SA), and abscisic acid (ABA) play a major role in adapting to stress. ABA in the early stages of fungal stress leads to the closure of the stomata to prevent the entry of pathogens. SA is another hormone that reciprocally triggers the MAPK signaling pathway during infection by biotrophic and hemi-biotrophic pathogens. Signals are transmitted from different domains of the receptors to SA, and subsequently SA activates multiple transcription factors such as TGA through a complex signaling network to express related PR genes. Expressed gene products will shape systemic acquired resistance (SAR) to improve host resistance against a wide range of pathogens. JA is another hormone that mainly plays a role in increasing resistance to necrotrophic pathogenic agents. The signaling pathway of this hormone together with ethylene synergistically participates in the expression of the plant defense-related gene *PDF1.2* [24,25,29,30].

PRR receptors can trigger reactive oxygen species (ROS) production and the PTI defense response by activating enzymes such as NADH oxidase and peroxidase. In this case, the toxicity caused by increasing the concentration of ROS will lead to the elimination of pathogens. On the other hand, the production of ROS initiates signaling pathways and thus causes other defense responses and cell death to prevent the spread of the pathogen to other parts. Activation of NADH oxidase (Figure 1e), also known as a respiratory burst oxidase homologs (RBOHs) enzyme, is achieved through phosphorylation by PRR associated BIK1 kinase to produce ROS in apoplastic space, which in turn can be transformed to H_2_O_2_ through peroxides’ catalytic activity. The membrane channel aquaporin (Figure 1f) has been shown to play a key role in transporting hydrogen peroxide produced in the apoplastic space into the cytosol and generating the corresponding defense responses. Cytosolic and apoplastic ROS burst can activate SAR, strengthen the cell wall and accumulate callus at the site of infection [31]. NLR receptors in the same manner as described above lead to ROS burst, which in turn triggers HR cell death. However, some compatible pathogens suppress the ROS burst by using effector proteins and cause host cell infection. Proteins such as lectin bind to the pectin in the cell wall of fungi and protect them from hydrolysis by enzymes that have accumulated as a result of PTI. In some cases, other proteins such as Mg1LysM and Mg3LysM from the fungus *Mycosphaerella graminicola* can bind to degrading enzymes in the host cell and block their activity [32]. In addition to the ROS burst caused by phosphorylation of the NADH oxidase in PRR and NLR receptors, the MAPK signaling pathway described earlier activates TFs such as WRKY and thus increases expression of genes involved in the production of apoplastic ROS (such NADP-ME and RBHO), leading to oxidative burst. Some pathogens can disrupt the kinase activity of these receptors and inhibit the activation of the MAPK and ROS signaling pathways by producing effectors. Suppression of these signaling pathways will ultimately inhibit PTI-derived defense responses and lead to the spread of disease in infected cells [33].

## 3. Experimental Approaches to Characterizing Disease Resistance Related Genes

Traditional genotyping methods using classical markers (such as RAPDs, RFLPs and AFLPs) entail unacceptable high costs, lack of automation and low efficiency as well as sometimes non-reproducible results [16]. In addition, these markers did not create a uniform coverage across genomes in mapping studies, and eventually were placed far away from the QTLs associated with the trait, which limited their usability in MAS. With the advent of NGS-based methods, it was possible to sequence the whole genome of plant and trees in a short time. Moreover, the study of transcriptomes and comprehensive studies of the expression patterns of different genes under various environmental conditions (e.g., biotic and abiotic stress conditions) has been made possible using these techniques. These methods, which include restriction site associated DNA sequencing (RAD-Seq), reduced representation libraries (RRLs), complexity reduction of polymorphic sequences (CRoPS), genotyping by sequencing (GBS) and multiplexed shotgun genotyping (MSG), were able to sequence simultaneously with the detection of markers, which resulted in the identification of a large number of polymorphic SNP markers [34]. Such markers have various advantages such as uniform distribution throughout the genome, low cost, and readiness of application, which enabled the genotyping of a large number of genotypes in a short time. The development of these markers will aid in accurate interval mapping and GWAS studies, identify QTLs associated with important traits on different chromosomes, and accelerate the breeding of forest trees through MAS. Since the abundance of SNP markers in forest trees is high (1 per 100 bp), it will facilitate the exploitation of genetic diversity in these trees in order to improve them for adapting to different environmental conditions [16]. There are only a few traits in forest trees whose inheritance follows Mendelian patterns. Most important traits, including growth, wood characteristics, resistance to diseases and pests, and abiotic stresses, are complex quantitative traits. While reverse genetic approaches such as genetic engineering can be considered as an alternative method to study such traits, the common application of such methods is hampered by limitations such as high cost and low-confidence transformation rates in most forest tree species [35]. In contrast, methods such as QTL mapping and association studies are widely used to investigate these traits. This technique is extremely important in reducing the time of cycle and cost of breeding programs of forest trees due to their long juvenile period.

### 3.1. QTL Mapping

QTL mapping has been used for more than three decades in studies on the complex traits of forest trees, including resistance to biotic and abiotic stresses and their developmental and wood properties [16]. The long time to reach the reproductive stage on the one hand and the problem of flowering on the other hand have made the length of the breeding period in forest trees a major problem [19]. On the other hand, there is no reliable correlation between the traits observed in young seedlings and mature trees, so direct selection based only on the traits observed in the juvenile stage cannot lead to the development of a variety with the desired traits [36]. Constructing genetic maps and QTL mapping have become important tools for marker-assisted selection (MAS). This strategy is especially important in trees, as it can lead to reliable early selection in young seedlings and thus increase gain per unit time and reduce the breeding period [37]. QTL mapping can help to understand the genetic basis of disease resistance variability in progeny derived from controlled crosses. However, segregating populations are often not available in non-domesticated and highly heterozygous species. Limitations such as long life cycle, inbreeding depression, and self-incompatibility also hamper the development of mapping populations (e.g., inbred lines and biparental populations) in open pollinated forest trees [38]. For this reason, F_1_ progenies derived from crosses between two heterozygous parents are common mapping populations for such species. In such populations, segregating test cross markers are analyzed separately for each parent. This strategy is known as “pseudo-testcross” since there is no prior information about the test cross marker, while in a traditional test cross the genotype of the tester is determined as homozygous recessive for a given locus [39,40,41].

Identification of QTLs and further study of genes within QTL intervals can also explain the molecular basis of diversity in different species in terms of resistance to disease. All this information can be helpful in adopting an effective management strategy for various diseases by selecting individuals with multiple resistance mechanisms in breeding programs [42]. Usually, QTLs derived from single biparental pedigrees do not exhibit similar patterns in different environmental conditions and genetic backgrounds. In order for the identified QTLs to be used in accurate genomic selection and MAS for other breeding populations, they must be validated over different years and locations and in other families [43]. Various molecular markers can be used to generate genetic maps. Classical molecular markers due to their low frequency, non-codominant characteristic, and lack of uniform distribution throughout the genome, led to the identification of few QTLs at long genomic distances. NGS-derived SNP markers are free of these weaknesses and can be used to develop high-density maps, e.g., [38,44]. These markers are developed by various NGS-based methods; among them, the GBS method is widely used in QTL mapping (reviewed in [34]). In the following, the QTL mapping method for finding QTLs related to disease resistance is briefly described. Initially, the mapping population is developed by crossing highly heterozygous parents. Markers showing heterozygosity in one or both parents will be screened in the progeny. After the onset of the disease by methods such as inoculation, the target population is evaluated based on the target traits such as scoring the disease, the rate of disease development in the area of infection, and growth related traits. The genotyping data are analyzed with the relevant software including JoinMap 4 [45] and MapMaker [46] and based on the recombination between the markers, the distances and the order of the markers are determined and the linkage map is constructed. Finally, using several methods such as linear regression analysis, multiple QTL, and composite interval mapping, the genotyping and phenotyping data are analyzed (e.g., with MapQTL 6), to identify putative QTLs and associated markers [47].

In the last two decades, several QTLs have been reported for resistance to various diseases in forest trees. These studies over the past decade have been largely based on NGS-based markers, including SNPs [37,42,43,48,49]. The use of QTL mapping in the eucalyptus tree has led to a deeper insight into the genetic architecture of the response to various diseases including rust, Calonectria leaf blight (CLB), and Ceratocystis wilt. The causative agent of rust disease in this tree is *Puccinia psidii*. A total of 4 QTLs (*Ppr2*, *Ppr3*, *Ppr4*, and *Ppr5*) were identified for this disease, placed on separate linkage groups. Two of these 4 QTLs (i.e., *Ppr4* and *Ppr5*) were effective in the hypersensitive reaction (HR) and the two other QTLs (i.e., *Ppr2* and *Ppr3*) were associated with the presence or absence of disease symptoms [42]. Because PTI-induced and basal responses result from structural barriers and chemical compounds such as waxy cuticle are symptomless, it can be concluded that the loci in *Ppr2*, *Ppr3* QTLs may be involved in such responses. On the other hand, it has been shown that QTLs underlying HR and their associated loci control the ETI response. A large number of genes were identified in the QTL confidence intervals, mainly belonging to the NB-LRR gene families. Proteins encoded by this gene family can directly or indirectly play a role in identifying effectors and triggering responses such as HR and asymptomatic immunity. Further, in these confidence intervals, several transcription factors were identified, including WRKY, bZIP and MYB families, which after activation at the end of different signal transduction pathways and entering the nucleus can lead to the expression of different defense genes [42]. In another study, in an F_1_ mapping of populations (*E. urophylla* S.T.Blake × *E. camaldulensis* Dehnh.) using microsatellite markers for Calonectria leaf blight (CLB) resistance, five putative QTLs were identified. Among these QTLs, resistance to defoliation 5 (*Rd5*) was identified as the major QTL, which accounted for about 50% of phenotypic variability. Validation of only one QTL (*Rd2*) out of 5 QTLs in two other unrelated populations indicates the interaction of QTLs with the genetic background, which is common in such heterogeneous populations. Therefore, the necessary caution should be taken in using such QTLs for MAS. To obtain more promising results, it is recommended to cover the whole genome using larger populations and high-throughput markers [48]. QTL analysis with SNP markers for resistance to myrtle rust (*Austropuccinia psidii*) and Quambalaria shoot blight (*Quambalaria pitereka*) in 360 F_1_ genotypes of Corymbia (*Corymbia torelliana* (F.Muell.) K.D.Hill & L.A.S.Johnson × *C. citriodora* (F.Muell.) A.R.Bean & M.W.McDonald) resulted in the identification of 22 QTLs. Of these 22 QTLs, 6 QTLs were related to rust and the rest were related to Quambalaria shoot blight (QBS) resistance. The two major groups of gene families within the confidence intervals of these QTLs were NBS-LRR and microRNAs (miRNAs). In addition to the role of miRNAs in regulating growth and development, hormonal activities and disease response, they also regulate NBS-LRR genes. Since QBS is a native disease for this tree, due to the mutual evolution of the host and the pathogen, the QTLs identified for resistance to this pathogen were of larger effect and independent from the QTLs for rust resistance. Therefore, most of the QTLs involved in QBS resistance were associated with the specificity of HR, which is carried out through specific *R* genes (oligogenic control) [50,51]. Past studies have shown that many of the QTLs identified for important diseases in eucalypts including CLB, Mycosphaerella leaf spot, and Ceratocystis wilt are mainly distributed on different linkage groups (LGs), indicating that resistance to these diseases is polygenic [43,48,52]. In *Eucalyptus*, chromosome 3 is one of the important chromosomes in which many syntenic genes of resistance to various fungi are located. It has been shown that this chromosome has undergone chromosomal changes during the evolutionary process, such as duplication of interchromosomal fragments, and on the other hand, it also contains conserved genes [43].

The GBS approach was used in backcross hybrids resulting from the crossing of American chestnut with two Chinese resistant cultivars (i.e., Mahogany and Nanking) to find QTLs associated with Phytophthora root rot (PRR) disease and to create saturated and high-density linkage maps [49]. A total of 17 QTLs were identified, which were distributed on 4 linkage groups related to parents. Since some crosses were phenotyped for two consecutive years, the results showed that environmental conditions could have a significant effect on the strength of identified QTLs. In general, the results showed that the genetic architecture of pathogen resistance in chestnut hybrids is similar to the pathogenic system between soybean and *Phytophthora soja*. Partial resistance in this plant is obtained by the action of multiple genes. It should be noted that in soybeans, resistance to *P. soja* (*Rps*) genes can configure race-specific resistance [53], while partial resistance in this plant is obtained by the action of multiple genes. These genes mainly have protected motifs such as LRR and NBS that are able to execute defense responses through the action of NBS-LRR receptors by activating the corresponding signaling pathways [49].

### 3.2. Association Mapping and Genome-Wide Association Studies (GWAS)

Linkage mapping studies are mainly performed in families with close relatives resulting from biparental crosses, which are based on recombination and segregation in this family to finally identify the genomic region controlling the desired trait (i.e., the QTL). Due to the limited recombination in such populations, the resulting maps do not have a high resolution and are not able to explain the diversity of complex traits [54,55,56]. However, it should be noted that in linkage mapping studies, controlling the possibility of recombination in the offspring is more practical than in association mapping (AM). In contrast, AM can be used in populations with a diverse set of individuals from natural populations, wild relatives, and other breeding populations. This method uses the linkage disequilibrium (LD) between markers and QTLs to identify the relationship between the trait and the marker. GWAS and candidate genes are two well-known methods for AM studies. All of these features imply the need to use AM studies in these populations in order to identify genomic regions controlling all traits showing diversity and causal genes by testing multiple alleles for associations in an experiment. LD analysis in GWAS uses not only recombination events after crossings, but also all recombination events that occurred between markers and genes over hundreds of generations. Involvement of multiple recombination events and testing of multiple alleles increases the resolution power of this method compared to linkage mapping [56]. QTL mapping in forest trees, for various reasons such as non-validation of QTLs, large tree genomes, impossibility of positional cloning of the identified QTLs, and their different expression across environments, has not been successful in dissecting the genetic architecture of highly complex traits. These limitations are overcome in GWAS by the application of a high marker density and unstructured natural populations. On the other hand, since the latter method is based on LD, if the trait-marker association is identified and validated, it is more reliable and likely to be located at close distance to functional genes. Since such a trait-marker association will remain for many generations and recombination events, GWAS derived markers in MAS are recommended [57].

In the last two decades, GWAS has been used as a key approach in assessing the natural diversity of crops, fruit and forest trees to explain the genetic architecture of various complex traits, including disease resistance [54,56,57]. In general, these studies have focused on traits such as wood characteristics, growth-related traits, resistance to diseases and pests, and abiotic stresses (Table 1). Using a large number of markers and evaluating an abundance of different germplasms, this method provides a suitable density of markers to evaluate the rate of LD decay and detect causative genes controlling traits of interest [56]. There are several factors that can affect the resolution of GWAS, which should be considered in the implementation of this method. Assessing phenotypic diversity, first the outliers should be removed from the raw data and then traits with high heritability estimates should be selected. At this stage, methods such as best linear unbiased estimator (BLUE) and best linear unbiased prediction (BLUP) are used to adjust the phenotypic scores based on the interactions of the genotype with the environment so that the power of GWAS is not negatively affected by this effect [55,58]. The number of individuals selected should also ensure sufficient genetic and phenotypic diversity in the study population. Populations with large numbers of individuals can increase the chances of identifying large quantities of the effect and also identifying QTLs with small phenotypic effects. Analysis of population structure with programs and methods such as STRUCTURE (using Q-matrix) [59], kinship matrix (K) and principal component analysis (PCA) is also one of the requirements of GWAS that is used to identify kinship between individuals and possible subgroups in the study population. Because individuals in GWAS do not have the same kinship, a false link between a trait and genotype can be detected if kinship is not considered. GWAS power also depends on the frequency of alleles, and it is possible that functional alleles with low abundance may not be detected despite their large effect on the phenotype. Rare alleles can be important in tree breeding, as some of them have major favorable phenotypic effects on complex traits such as resistance to certain diseases [19,60]. Selection of wild relatives and landraces and efficient genotyping methods such as GBS can help increase the power of GWAS to identify these alleles. Finally, calculating LD through statistics such as r^2^ and D’ is also necessary to estimate the distances between loci and identify common mutation and recombination history between them. The calculated LD is shown as r^2^ and D’ in a plot called the LD decay plot for each chromosome. In this plot, the genetic distance is displayed on the X-axis and the calculated LD values (as r^2^) are displayed on the Y-axis. Finally, a nonlinear logarithmic regression curve for the scatter points is obtained. In AM studies, a threshold for LD is usually considered, and SNPs with r^2^ ˃ 0.2 are considered for statistical analysis. The rate of LD decay over distances determines the mapping resolution, while LD decay at short distances leads to high-resolution maps and vice versa. All factors resulting in deviation from Hardy–Weinberg equilibrium, including migration, genetic drift, non-random crossover, selection, and mutation, can affect LD. The LD value is also used to determine the number of sufficient markers to cover the entire genome. If LD between two significant SNPs is small, it is likely that these two SNPs are independent of each other and are located in a separate QTL, and vice versa.

False positive identification is one of the main limitations of GWAS that should be considered, although appropriate models which include affecting covariates such as PCA, population structure (Q matrix) and kinship (K), can also impede such identifications. In addition, genetic engineering methods (e.g., gene silencing) can be useful for validation of the identified associations. However, it should be noted that in many tree species these methods have not been established [35,56,61].

In the following, the GWAS procedure in plant resistance to a variety of diseases is briefly explained. The first step involves selecting the right number of individuals for the population to be studied. In the case of forest trees, groups of natural individuals are usually selected without the use of controlled crosses with unknown ancestors to take advantage of natural recombination events that have occurred during their evolution. Selected individuals are exposed to the target disease by using artificial inoculation and then phenotyped based on the target trait or traits. Usually, these analyses are performed in replicated provenance or progeny trials using a randomized design (for example, randomized complete block design) to minimize environmental effects on the phenotype. In this case, based on the observed disease symptoms, a diverse range of disease responses from asymptomatic to the presence of obvious symptoms in different tissues are usually scored and raw data are obtained, the outliers removed and the adjusted phenotype values calculated using BLUP or BLUE [58,62]. The broad-sense heritability is also usually calculated for raw data to determine the genetic portion in controlling traits. Phenotypic measurements must be repeated in different years and places, which is a major challenge due to the large number of individuals used in this method. In the next step, individuals are genotyped using suitable markers. The reason for the use of SNPs, which are usually obtained with the GBS method, is their wide distribution in the genome, low cost, and high reproducibility. Population structure determined using PCA, the software STRUCTURE, and a kinship matrix (K) are included in appropriate GWAS models to detect valid marker-trait associations [55,63]. Finally, using models such as general linear models (GLMs) and mixed linear models (MLMs), GWAS analysis is performed and the relationships between the target trait and the alleles are identified. Significant associations of each trait with each SNP are assessed using Bonferroni correction (BC) and false discovery rate (FDR). In general, the FDR method has a high degree of flexibility and is less conservative than the BC method, and is recommended for determining independent associations of traits and markers. GWAS outputs may depend on the type of software used and can include various parameters and graphs (Figure 2).

**Table 1 ijms-23-12315-t001:** Several significant marker-trait (SNP and DArT) associations found by GWAS models in different forest tree species.

Species	Target Trait	Number of Samples	Number of Markers	Significant Markers	GWAS Model	References
*Cornus florida* L.	Resistance to dogwood anthracnose and ecological pressures	289	3134	14–29	Latent factor mixed modeling	[64]
*Eucalyptus*	Growth and wood property traits	1123	37,832	78	FarmCPU	[65]
*Eucalyptus*	DBH, diameter at breast height; HT, total height.	3373	41,320	11–3	linear model-based association (LMA) and mixed linear model-based association (MLMA)	[66]
*E. globulus*	Growth and wood property traits	303	2,364 DArT	18	Unified Mixed Model	[67]
*E. grandis*	Terpene traits, 1,8-cineole, c-terpinene, and p-cymene	416	15,387	21–32 and 28	Efficient Mixed-Model Association eXpedited (EMMAX)	[68]
*Fagus* *grandifolia Ehrh.*	Beech bark disease (BBD)	514	16,000	4	logistic regression model	[69]
*Hevea brasiliensis*	Latex yield	170	21,146	4	MLMA	[70]
*Picea abies* (L.) H. Karst.	Susceptibility to H. parviporum	533	373,384	36	EMMAX	[71]
*P. abies* L. Karst	Heterobasidion rot caused by H. annosum s.s.	400	63,760	12	MLM	[72]
*P. abies L. Karst*	Resistance to H. parviporum	466	197,399	11	LASSO	[73]
*P. crassifolia Kom.*	Earlywood tracheid traits	106	12,275,765	96	MLM	[74]
*Pinus contorta*	Serotiny	98	95,000	11	Bayesian generalized linear model (GLM)	[75]
*P. taeda*	Solid wood and wood chemistry traits	435	58	4	MLM	[76]
*P. taeda*	Carbon isotope discrimination	961	46	4	MLM	[77]
*P. taeda*	Pitch canker resistance	498	7216	10	Bayesian association with missing data (BAMD)	[78]
*Populus trichocarpa*	16 wood chemistry/ ultrastructure traits	334	29,233	141	regression model	[79]
*P. trichocarpa*	Resistance to *Sphaerulina musiva*	3404	8,253,066	96	EMMA in EMMAX software	[80]
*P.trichocarpa*	Stomatal traits	2447	34,131	25	GLM	[81]

Identifying important alleles and supporting the introgression of these alleles from diverse individuals into selected populations is one of the important applications of AM. In this regard, limited studies have been conducted in forest trees to identify disease resistance related alleles. Discovering markers located at close distances to QTLs controlling the target trait can also be another practical aspect of AM. These markers can be used to identify individuals and genotypes of interest instead of directly selecting them based on the trait. This strategy is known as MAS, which can significantly reduce the breeding program process, especially in perennial trees that have a long juvenile period. After identifying the important and significant QTLs to which the significant markers are attached, these QTLs can be used to find their physical position and identify candidate genes [64]. Identification of these genes in disease resistance studies could reveal the genetic architecture of the response to these diseases. From another point of view, cloning and overexpression of these genes in the same plant or its expression in another species can be a step in the development of resistant plants to various diseases [3,35].

The use of regional heritability mapping (RHM) along with GWAS can help identify loci containing rare allelic effects that contribute little to the phenotypic variance. Usually, the GWAS method is not able to estimate this variance due to insufficient LD between these sites and the SNPs used. RHM uses both common and rare SNP variants found in short intervals of the genome to create a genomic relationship matrix (GRM) which finally will be used for calculating phenotypic variance related to such segments. This method is able to identify more QTLs with a lower false positive rate than the GWAS method [82]. GWAS and RHM were used to find loci associated to growth, wood and disease resistance traits in 768 eucalyptus hybrid trees resulting from a *E. grandis* L. x *E. urophylla* cross. According to the –log10 *p*-value and permutation threshold, the association between *Puccinia psidii* (also known as *Austropuccinia psidii* Beenken) rust disease resistance and the corresponding SNP (placed on chromosome 3) was identified as the most significant association. RHM led to the identification of 4 QTLs for resistance to the disease, which, along with two other QTLs associated with wood quality traits, were located on the same chromosome [83]. The first major QTL for resistance (i.e., *Ppr1*) to the disease was identified by Junghans et al. [84], and subsequent further studies led to its validation and positioning on chromosome 3 [85,86]. The results showed that the RHM method compared to GWAS leads to a 3-fold increase in heritability for various traits, including disease resistance, which indicates the superiority of the RHM method in identifying the association of loci with complex traits. This superiority can be explained by the fact that RHM using the combined effect of several SNP variants increases the likelihood of association in comparison with GWAS that only uses single marker effects for association analyses [82,83].

Recently, in another study, 33 significant SNPs were identified for resistance to myrtle rust disease in the *E. oblique* L’Hér. tree. Most of the significant markers were associated with the presence or absence of symptoms of the disease, hypersensitivity and pustules. Five of these markers were located on each of chromosomes 3 and 4, and the rest were on the other chromosomes. Several genes (i.e., 67) identified around these markers (±5 kb of each SNP) indicate that resistance to the disease includes a wide range of responses, including PTI and basal resistance (asymptotic responses) to ETI (e.g., cell death and HR) [58]. The results of this study and other studies have shown that resistance to this disease in eucalyptus and other related trees is controlled by diverse loci with different effects, indicating the complexity of this trait [42,87,88]. Previous studies in *Eucalyptus* identified not only *NBR-LRR* genes, but also some other groups of *R* genes including kinases, hydrolases, and nucleotide-binding proteins and transcription factors (e.g., WRKY and bHLH) to be involved in resistance against myrtle rust. The presence of different *R* gene families in plants and trees allows effective identification of pathogenic effectors that change over time [42,58,89]. In previous studies, many SNPs of resistance to this disease and other fungal diseases were mainly located on chromosome 3 [43,52,89,90]. In a recent study, several of the significant SNPs were also located on the same chromosome, which indicates the presence of conserved genomic regions, gene clusters, and pleiotropic genes for resistance to this disease and other diseases on this chromosome [58].

Although AM-based methods can overcome most of the limitations of the linkage mapping method, this method itself also has limitations that restrict its practical use in MAS. Among the limitations of this method, the following can be mentioned: low phenotypic variance explained by identified associations, limited LD, false positive/negatives, and rare alleles.

### 3.3. Genomic Selection

With the advancement of high-throughput genotyping methods, the genomic selection (GS) procedure has become an alternative paradigm for MAS. This method is of considerable importance in the study of complex quantitative traits with low heritability (such as yield, disease resistance, and wood quality) that are affected by environment and expensive to measure and appear after a long time of growth. Early selection of elite genotypes using this method not only reduces the cost and amount of work of breeding programs, but also makes it possible to identify successful parental crossings that ultimately lead to desired allelic combinations in the progeny [19,67,91,92]. The GS method, unlike MAS, uses the data of all markers without considering the significance of the marker-trait relationship, and the magnitude of the QTL effect [93]. In this method, a suitable training population is selected and genotyped using genome-wide markers, while phenotypes are scored in different places and years. The obtained phenotypic and genotypic data are used to fit a prediction model and estimate its parameters that will be used later in the breeding population. The fitted model is cross-validated in a validation population that is genetically related to the training population. The breeding population is also genotyped with the same number of markers (no need for phenotyping). Using the genotyping data of the breeding population and with the help of the marker effect on the trait obtained from the training model, the genomic estimated breeding values (GEBVs) of the individuals are calculated. Finally, the individuals of the population are selected based on the calculated GEBVs [66]. In general, several methods for calculating GEBV values are based on Bayesian (e.g., Bayesian A, B, Bayesian ridge regression and Bayesian least absolute shrinkage and selection operator) and BLUP (genomic and ridge regression) methods. The application of different methods depends on the conditions of the research in question, for example, when there is less kinship between the breeding and training population, the Bayesian method is preferred to GBLUP, and on the other hand, for traits having high heritability and traits that are controlled by a moderate number of genes, it is better to use the Bayesian A and B method, respectively [92]. Since GEBVs can be estimated in seedlings, the GS approach can play an essential role in increasing genetic gain per time unit and reducing the breeding period in forest trees [94].

In the last decade, GS has attracted the attention of many forest tree breeders in order to identify superior genotypes for various traits. SNP and DArT genotyping platforms available for *Eucalyptus* have made it possible to accurately estimate genetic parameters and construct pedigree architecture; therefore, most GS studies in forest trees have been performed in eucalypts [95]. However, recently, due to the emergence of NGS sequencing methods and the possibility of discovering SNPs in different species, GS has also been started in other forest trees. In a recent study on the nut yield of *Macadamia integrifolia* Maiden and Betche, moderate and high prediction accuracy was obtained for yield and yield stability traits, respectively. The results of this study showed that GS can reduce the selection cycle for yield by seven years (from 21 to 14 years) and double the genetic gain for this trait [94]. Some reports showed the predictive accuracy of traits related to the growth and quality of wood in trees such as fire tree, *Pinus pinaster* Ait. and *Eucalyptus pellita* F. Muell to be moderate [95,96,97]. High prediction accuracy increases the confidence level in selecting the top individuals and depends on various factors such as population size, nature of a trait, model type, number of markers used and the amount of LD [94]. Preliminary studies regarding the evaluation of factors affecting the GS accuracy in forest trees have shown that among them, LD between marker-trait and heritability, and the number of QTLs have the greatest and least impact on the accuracy of GS, respectively. On the other hand, for the successful application of GS in forest trees, a sufficient number of training populations and a high marker density should be used. For a population with an effective size of ˂30, the density should be 2-3 markers per centimorgan, and for larger populations, this value should be 20 to have 50% prediction accuracy even for traits with low heritability controlled by 100 QTLs [91]. Unlike cultivated and self-pollinated plant populations, in most forest tree species with large effective population sizes and an open pollination system, LD has a significant decrease in short distances, and in order to make an accurate prediction in these species, a high marker density should be used [95,98]. In addition, close kinship of the individuals of the training and validation population can lead to an increase in the prediction accuracy of GS models [99].

More recently, GS has also been used for disease resistance traits (Table 2). These traits are often controlled by oligo QTLs that explain a significant portion of the phenotypic variance. These features guarantee the success of MAS in such traits and on the other hand increase their predictability through GS [100]. Different GS methods (such as Bayes A, Cπ Bayesian LASSO and RR–BLUP) were used for several traits in *Pinus taeda* L., including growth, development, wood quality, and fusiform rust resistance, in order to evaluate the efficiency and accuracy of their estimates. The lowest (0.17) and the highest (0.51) predictability of different methods was related to lignin and branch angle traits, respectively. Bayes A and Cπ, compared to other methods, showed better predictive power in relation to fusiform rust resistance. This outcome is related to the genetic architecture of traits and shows the high capability of these methods in predicting a trait with simple genetic architecture controlled by a few loci with large effect. The poor performance of the RR-BLUP in oligogenic traits is due to fitting a large number of markers in modeling the phenotypic variation of traits that are controlled by a small number of loci. In contrast, in Bayes methods the effects of each marker are considered specifically and they are not treated equally as in the BLUP method [100,101]. Using the RR-BLUP method with adding individual makers to the model and calculating the predictive ability, it was determined that the maximum predictive ability is achieved with 110 markers for the resistance traits in *P. taeda*. However, the number of markers required for most traits related to growth, development and wood quality to reach the maximum predictive ability was over 600 [101]. The RR-BLUP B method with the necessary number of markers to reach maximum predictive ability performed better than Bayes and traditional RR-BLUP methods [101]. The predictive ability of SNP markers for Fusarium blight resistance in *P. taeda* is affected by the type of model used and the cross validation scenarios. Higher predictability is obtained when the relationship between training and validation populations is greater. For this reason, the random selection of samples from the population and creation of training and validation populations compared to random sampling scenarios of families and within families leads to the highest prediction value (0.71–0.76) of different models. In different cross validation scenarios, the Bayesian method is preferred over the whole genome regression models in order to capture major effect QTLs involved in Fusarium rust resistance [102].

Knowing the genetic architecture of a trait can help in adopting the appropriate model by including additive or additive-dominance effects in the model to increase the predictive ability. For polygenic traits such as height, which are controlled by a large number of genes (dominance effects), application of additive-dominance models leads to increased prediction accuracy. On the other hand, additive models work better to achieve high prediction accuracy regarding oligogenic traits such as rust resistance for which dominance effects are negligible. Usually, additive-dominance models perform better than the additive models alone for oligogenic traits with high dominance effects. Prediction accuracy of additive-dominance models decreases with the progress of generations, however. Therefore, these models are more practical in species that are propagated through vegetative methods [103,104].

It has been found that the selection of a subset of top significant SNPs based on GWAS *p* values can create more accurate predictive models than randomly selected SNPs of the same subsets. Highest prediction accuracy (0.67) for ash die back (ADB) disease resistance was achieved by the top 10,000 significant SNPs of a training population (out of 9,347,243 SNPs) which were selected from GWAS. In this study, using 200 informative SNPs in the test population, the fitted model was able to assign trees with the highest resistance to this disease to healthy groups with a prediction accuracy of 0.9. Since the model was fitted on pooled sequencing data (31 pools from 1250 trees), the obtained model can be used with high accuracy in evaluating the resistance to ADB of different populations of ash trees that are genetically different [105].

The process of forest restoration of different tree species can take more than a century and the GS method can be used in such programs and accelerate this process. The restoration of the American chestnut forests, which were destroyed by blight, started in the 1920s [106]. More recently, promising results have been obtained using MAS and GS studies. Recently, GS has been used to accelerate the selection of resistant trees in two chestnut backcross populations (BC_3_F_2_) named “Clapper” and “Graves”. The genes of resistance to this disease in these populations was descended from two different *Castanea mollissima* Blume Chinese donor parents. The fitted models (HBLUP and Bayes C) performed similarly but their capabilities were better than pedigree methods in the selection of 1% of the individuals for canker severity and late-developing blight traits in BC_3_F_2_ selection candidates and BC_3_F_3_ (maximum ryŷ = 0.64 for “Clapper” and ryŷ = 0.48 for “Graves”) progeny, respectively [107]. This capability can be attributed to the more accurate and higher estimation of individual relatedness in the GS compared to the pedigree method [108]. The results of this study showed that resistance to this disease is a polygenic trait with a low heritability and the proportion of the genome inherited from American chestnut in individuals has a negative relationship with resistance. Therefore, according to the above facts, to improve resistance to this disease, a breeding program including recurrent selection, backcrossing with a lower number of generations and using new sources of resistance, evaluation of resistance in the backcross trees and the progeny of the selected individuals will have promising results [107].

**Table 2 ijms-23-12315-t002:** GS studies in forest tree breeding applied for disease resistance.

Species	Target Trait	Population Size and Type	Number of Markers	Prediction Accuracy	Prediction Model	References
*Pinus taeda* L.	Growth, wood quality, and fusiform rust resistance	921 from 61 full-sib families	4,853 SNPs	0.17-0.51	Bayes A, Cπ Bayesian LASSO and RR–BLUP	[101]
*P. taeda* L.	Fusiform rust resistance	721 from 5 full-sib families	16,920 SNPs	0.71–0.76	Ridge regression, Bayes B, and Bayes Cπ	[102]
*Fraxinus excelsior*	Resistance to ash dieback disease	1,250 from 15 native seed zones	10,000 SNPs	0.67	RR–BLUP	[105]
*Castanea dentata*	Mean canker severity	Progeny of 1,230 from BC_3_F_2_	71,507 SNPs	0.06-0.30	HBLUP, ABLUP, and Bayes C	[107]
*Theobroma cacao*	Resistance to black pod rot (BPR), witches’ broom disease (WBD), and frosty pod rot disease (FPRD) diseases	1,345 accessions from F_1_ progeny	9,640 SNPs	0.065-0.478	G-BLUP	[109]

## 4. Future Application of GWAS and GS

With expanding access to whole genome sequences of forest tree species and the possibility of discovering high-throughput markers (e.g., SNPs), the use of GWAS and GS for the breeding of forest trees is increasing. For the successful use of this method in breeding programs, it is necessary to develop and improve effective statistical models and software. Since the phenotyping step is an essential step in such studies, the use of precise phenotyping tools and methods is another challenge that must be considered in the evaluation of individuals and populations. On the other hand, to use the significant markers discovered in future experiments, trait-marker associations should be validated in other populations and in repeated experiments in different places. GWAS in future forest tree breeding for disease resistance also should take into account the climate change issue, since some weather conditions such as extreme drought can intensify the severity of a disease and increase the susceptibility and mortality of genotypes that appear resistant under favorable conditions. Therefore, strategies such as simultaneous selection for disease and drought resistance alleles should be considered in forest tree breeding programs. Using the information for all the markers, the GS procedure can be applied as an alternative or a complementary method for MAS not only for wood quality and growth related traits but also for disease resistance traits. Knowing the type of trait (simple or complex inheritance) is very important in choosing a statistical method and the development of new statistical methods according to the degree of complexity of the target traits can increase the efficiency of GS. Breeding of forest trees by using MAS and GS for disease resistance is not considered as a common strategy in forest management programs, especially in developing countries. However, it is expected that due to the significant diversity in such natural populations and by reducing the cost of whole genome resequencing, GS will become more important in the future. Methods such as GWAS with an indirect selection approach through MAS and GS can lead to the development of elite parents or genotypes that can be used in the future to create desired populations. These populations can finally be propagated and used in an integrated form with other management methods and gradually replace sensitive trees. In addition, by identifying resistance genes and overexpressing or introducing them into other forest tree species with genetic engineering methods or editing specific genomic regions using new techniques such as CRISPER/Cas, the development of transgenic forest trees resistant to various diseases is not far from reach. In this regard, the non-response of many forest tree species to tissue culture and gene transformation methods is a serious challenge that requires much work in the future so that the methods mentioned above can be used.

## 5. Conclusions

In recent years, with the help of NGS methods, genomics and transcriptomics studies in forest trees have increased significantly and have helped the breeding programs of these trees. However, most of these studies have been conducted on economically valuable trees, and ecologically valuable species have been neglected. With the advent of this technology and availability of whole genome sequences, high-throughput markers (e.g., SNPs) were also discovered, which enable quick and cheaper genotyping of different species. Due to their uniform distribution throughout the genome, these markers enabled the construction of high-density linkage maps and the identification of QTLs related to resistance traits to various diseases. Due to the limitations in the interval mapping method and the unique characteristics of forest trees, the AM method has attracted the attention of researchers in this field. Both of these methods are able to dissect the architecture of resistance to diseases and the different genes involved. Identifying markers related to disease resistance traits using these methods can increase forest tree improvement programs through MAS. Along with other challenges of AM and interval mapping methods, paying attention to QTL effects and marker-trait associations is one of the major disadvantages of these methods, and ignores valuable genome information. The GS method, using the information on all markers, is the best method to greatly increase the speed of breeding in complex traits related to disease resistance. However, the number of such studies in forest trees is limited due to various reasons such as the lack of reference genomes; the development of databases is limited to certain trees. Choosing the best model and using appropriate sampling methods to ensure the highest relatedness between the training and verification populations has a significant impact on the success of GS studies. AM and interval mapping (for MAS) and GS can be used according to the type of target trait and increase the breeding gain per year of perennial forest trees. However, the success of these types of studies depends on the development of accurate statistical models and accurate phenotyping tools and methods.

## Figures and Tables

**Figure 1 ijms-23-12315-f001:**
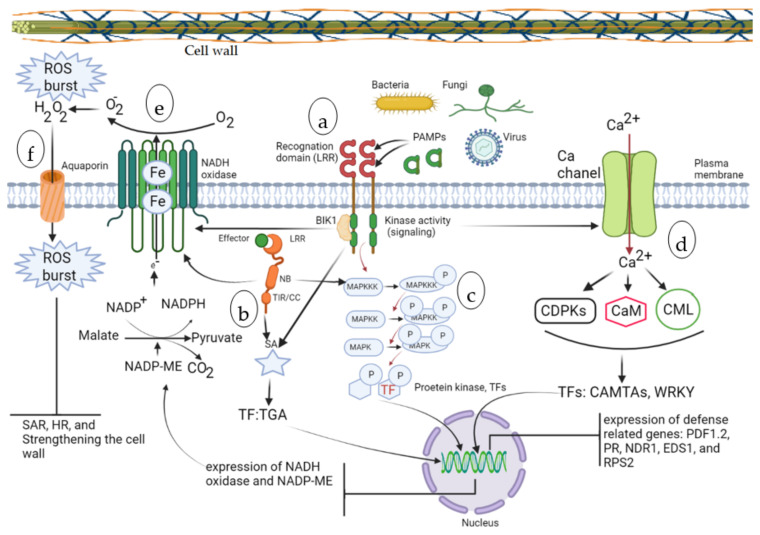
Schematic representation of pathogen perception, different signaling pathways, and corresponding defense responses in trees. (**a**) pattern recognition receptors (PRRs), (**b**) nucleotide binding site-leucine-rich repeats (NBS-LRR) family of receptors, (**c**) mitogen-activated protein kinase (MAPK) cascade, (**d**) turbulence in calcium concentration during stress and its perception by several proteins, (**e**) activation of NADH oxidase and production of reactive oxygen species (ROS), (**f**) transport of hydrogen peroxide through an aquaporin channel.

**Figure 2 ijms-23-12315-f002:**
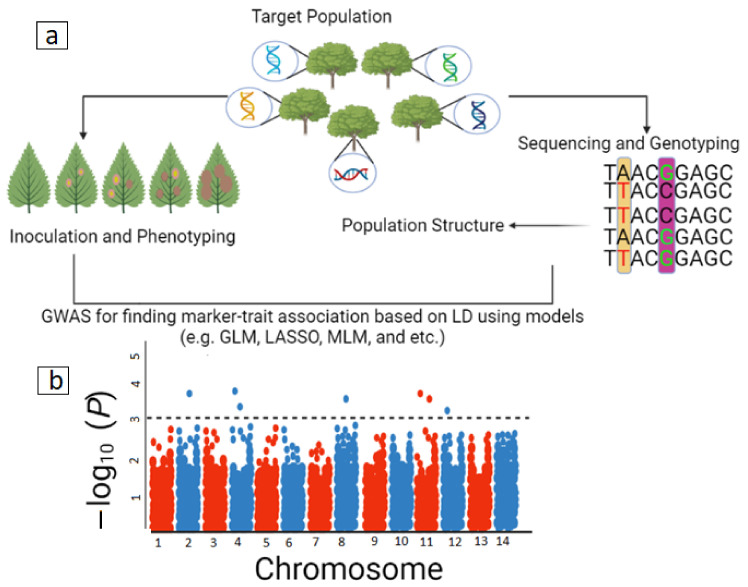
(**a**) Schematic representation of GWAS and (**b**) resulting Manhattan plot in forest trees for finding marker-trait associations in disease resistance studies. On the *x*-axis of this plot, the markers are displayed on different chromosomes and on the *y*-axis, the –log10 *p*-values related to each marker are displayed. A threshold line is considered for p-values, and the SNP markers located above this line can be considered as potential QTL (modified from [55]).

## Data Availability

Not applicable.

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
