# Peer review of "Genome-Wide SNP Markers Accelerate Perennial Forest Tree Breeding Rate for Disease Resistance through Marker-Assisted and Genome-Wide Selection"

_ijms, 2022, doi:10.3390/ijms232012315_

Round 1

Reviewer 1 Report

Comments to the Authors

In the current review manuscript entitled “Genome-Wide SNP Markers Accelerate Perennial Forest Tree Breeding Rate for Disease Resistance through Marker-Assisted Selection and Genome-Wide Selection,”. The authors' main objective was highlighting the latest advanced SNP marker application in forest tree breeding. These markers are used in QTL mapping, and GWAS approaches for identifying significant markers associated with disease resistance and developing a prediction model in genomic selection studies for future breeding programs to screen breeding populations.  The manuscript is well-written, organized, and prepared. In conclusion, if my suggestions can be addressed successfully in a revision, I believe the manuscript can be published in this journal.

It is better to summarize the manuscript and add your perspective and suggestions for future studies at the end.

Line 97; write the full phrase for QTL (quantitative trait locus), then use the abbreviation. 

Line 110. Use the abbreviation for next-generation sequencing as explained previous paragraph. 

 Line 112. write out the full phrase for SNP, then use the abbreviation.

Line 119. Use the abbreviation for genome-wide association studies as explained in the previous paragraph. 

Line 126. their? Rewrite the sentence.

Line 171. Full phrase of MAPK?

Line 231. Add more details for the figure by dividing it into several parts, including a, b, c, d, and e, and refer to these parts in appropriate sections of the manuscript text. 

Line 317. “Eucalyptus” use the italic format if you use a name for the genus of an organism.

Line 380. (i.e., QTL)?

Line 403. Replace GWAD with GWAS.

Line 415. Write out the full phrases for “BLUE and BLUP.”

Line 663. “BC3F2” apply subscript format for numbers

Line 667. “BC3F2” use subscript format for numbers

668. “BC3F3” use subscript format for numbers

Reviewer 2 Report

This manuscript is a review focusing on genome-wide markers for tree breeding. While the topic is interesting, I think that several points should be improved.

A section with perspectives for future work should be added.

More figures or tables could be added.

The section “Molecular mechanisms of tree responses to diseases” contains mostly very general defense mechanisms. It would be interesting to develop tree-specific molecular mechanisms, with the literature associated.
